# Muscle Hypertrophy Is Linked to Changes in the Oxidative and Proteolytic Systems during Early Tenderization of the Spanish Breed “Asturiana de los Valles”

**DOI:** 10.3390/foods13030443

**Published:** 2024-01-30

**Authors:** Marina García-Macia, Verónica Sierra, Adrián Santos-Ledo, Beatriz de Luxán-Delgado, Yaiza Potes-Ochoa, Susana Rodríguez-González, Mamen Oliván, Ana Coto-Montes

**Affiliations:** 1Department of Biochemistry and Molecular Biology, University of Salamanca, 37007 Salamanca, Spain; 2Institute of Functional Biology and Genomics (IBFG), University of Salamanca/CSIC, 37007 Salamanca, Spain; 3Institute of Biomedical Research of Salamanca (IBSAL), University Hospital of Salamanca, 37007 Salamanca, Spain; 4Centre for Biomedical Investigations Network on Frailty and Ageing (CIBERFES), 28029 Madrid, Spain; 5Servicio Regional de Investigación y Desarrollo Agroalimentario (SERIDA), 33300 Villaviciosa, Spain; veroniss@serida.org (V.S.); mcolivan@serida.org (M.O.); 6Instituto de Neurociencias de Castilla y León (INCyL), 37007 Salamanca, Spain; santosledo@usal.es; 7Department of Human Anatomy and Histology, University of Salamanca, 37007 Salamanca, Spain; 8Health Research Institute of Asturias (ISPA), 33011 Oviedo, Spain; b.luxandelgado@gmail.com (B.d.L.-D.); potesyaiza@uniovi.es (Y.P.-O.); acoto@uniovi.es (A.C.-M.); 9Instituto Universitario de Oncología del Principado de Asturias (IUOPA), 33006 Oviedo, Spain; 10Departamento de Morfología y Biología Celular, Facultad de Medicina, Universidad de Oviedo, 33003 Oviedo, Spain; rodriguezgsusana@gmail.com

**Keywords:** muscular hypertrophy, lipid oxidation, protein oxidation, total antioxidant activity, proteolysis, autophagy, meat aging, beef, Asturiana de los Valles

## Abstract

For fresh meat consumers, eating satisfaction is of utmost importance and tenderness is one of the most important characteristics in this regard. Our study examined beef of different animal biotypes of the autochthonous breed “Asturiana de los Valles” (AV) to determine if early postmortem oxidative and proteolytic processes may influence the final tenderness of the product. This meat-specialized breed shows different biotypes depending on the frequency of a myostatin mutation “*mh*” that induces double-muscling or muscular hypertrophy (*mh*/*mh*, *mh*/+, +/+). Samples from the longissimus dorsi muscles of yearling bulls were analyzed during the first 24 h postmortem. Changes in the redox balance of muscle cells were significant in the first hours after slaughter; total antioxidant activity was higher in the *mh*/*mh* biotype and it followed the shortening of the sarcomeres, a key parameter in understanding meat tenderness. The two proteolytic systems studied (proteasome and lysosome) followed distinct patterns. Proteasome activity was higher in the (*mh*/+) biotype, which correlated with higher protein damage. Lysosome proteolysis was increased in the more tender biotypes (*mh* genotypes). Autophagic activation showed significant differences between the biotypes, with (*mh*/*mh*) showing more intense basal autophagy at the beginning of the postmortem period that decreased gradually (*p* < 0.001), while in the normal biotype (+*/*+), it was slightly delayed and then increased progressively (*p* < 0.001). These results suggest that this type of catalytic process and antioxidant activity could contribute to the earlier disintegration of the myofibers, particularly in the *mh/mh* biotypes, and influence the conversion of muscle into meat.

## 1. Introduction

Fresh meat consumers value different components of beef palatability, including flavor and juiciness, but always highlight tenderness [1,2]. This quality is key when consumers choose a specific brand and impacts the economic benefits of the livestock production sector. Cattle intrinsic factors, such as breed and genetics, have a decisive influence on meat quality. The protected geographical indication (PGI) quality label “Ternera Asturiana” from Spain is considered one of the best meats in Europe [3] and covers beef of different biotypes of the local beef breed “Asturiana de los Valles” (AV), which depends on the frequency of double-muscling phenomenon or muscle hypertrophy “*mh*” genotype (*mh*/*mh*, *mh*/+, +/+). A previous study from our group indicated that the AV biotype directly impacts postmortem tenderization rates, being faster in *mh* genotypes (*mh*/*mh*, *mh*/+), while longer aging times are required in meat from normal (+*/*+) AV animals [4].

Meat aging is one of the most important processes to obtain a satisfactory degree of tenderness. Upon slaughter and exsanguination, the first stimulus that muscle cells detect is the interruption of blood flow. Consequently, muscle cells are irreversibly deprived of nutrients and oxygen. Low levels of oxygen inhibit the mitochondrial electron transport chain, which results in redox changes and raises the production of reactive oxygen substances (ROS) that produce oxidative stress, which in turn target proteins and lipids [5,6]. In response to this oxidative challenge, cells trigger different pro-proteolytic pathways, such as the calpain system (previously assessed by our group [2]), the proteasome, or lysosome-related proteolysis (autophagy). Postmortem muscle proteolysis varies among animal species and among different muscle types. However, most of the studies have been performed after rigor mortis and little is known about ROS during the first hours after slaughter when these molecules are at their highest levels [7]. Furthermore, it has been shown that the degree of protein oxidation modulates the quality of meat products [5]. Also, the rate and extent of postmortem proteolysis could be responsible for different tenderization rates observed among different beef breeds or genotypes [8,9]. Our research group described autophagy as an important cell process triggered after slaughter with significant consequences on muscle destruction and meat tenderization [10]. However, whether autophagy activity changes due to *mh* genotypes during meat aging is unknown. This knowledge will help us to better understand meat tenderization and thus increase customer satisfaction.

The aim of this work was to study for first time the effect of the postmortem evolution of the muscle oxidative balance (Total Antioxidant Activity (TAA), lipid, and protein oxidation), the main proteolytic systems (proteasome and autophagy), and sarcomere shortening (as a putative tenderness indicator) on bovine longissimus dorsi muscle from different biotypes protected by the PGI “Ternera Asturiana” (AV (*mh*/*mh*), AV (*mh*/+) and AV (+/+)).

## 2. Materials and Methods

### 2.1. Animals and Sampling Procedure

Eighteen yearling bulls of a local breed from northern Spain, AV, were studied. This is a meat-specialized breed and presents a myostatin mutation (*mh*) that induces double-muscling characteristics, giving rise to three different biotypes: homozygous (*mh*/*mh*), heterozygous (*mh*/+), and normal (+/+).

After 90 days of age, blood samples from AV animals were obtained and analyzed to determine the presence of the 11-bp deletion in the coding sequence of the myostatin gene, which causes double-muscling in cattle [11]. Calves were managed with their mothers from birth (winter) to weaning (October), then fed on concentrate (84% barley meal, 10% soya meal, 3% fat, 3% minerals, vitamins, and oligoelements) and barley straw ad libitum. Animals were slaughtered at 18 months of age in a commercial abattoir following approved EU procedures. After slaughtering and dressing, the carcasses were transferred to a cold room at 3 °C within 1 h after slaughter.

For the study of oxidative stress damage and proteolytic activities in muscle extracts during the early postmortem period, 6 animals from each biotype (AV (*mh*/*mh*), AV (*mh*/+), and AV (+/+)) were selected. All samples were protected by the PGI “Ternera Asturiana”.

From these 18 carcasses, 20 g muscle samples were taken from the longissimus dorsi at 2, 12, and 24 h postmortem. These samples were immediately frozen in liquid nitrogen and stored at −80 °C until analyzed. It is assumed that muscle samples taken at 2 h postmortem represent the basal levels for oxidative and autophagic markers [11]. This was the shortest time postmortem at which carcass tissues could be extracted due to the standard commercial procedures. The pH and temperature of the muscle at 2 h postmortem were similar to those found in living tissue [12].

### 2.2. Tissue Extraction

A total of 1 g of muscle tissue per animal and aging time (2, 12, and 24 h postmortem) was thawed and homogenized using an Ultra-Turrax T25 (JANK & KUNKEL IKA-Labortechnik, Staufen im Breisgau, Germany) in 9 mL of homogenization buffer (10 mM potassium phosphate, pH 7.4, containing 50 mM sodium chloride and 0.1% Triton-X 100). The tissue homogenates were then centrifuged for 6 min at 1000× *g* at 4 °C, and the supernatants were collected and then centrifuged under the same conditions. The supernatant was collected, and these sample extracts were used in Western blotting for autophagic markers analyses, as well as for enzyme activity analyses [10].

Supernatant protein content was quantified by the Bradford method [13] at 595 nm using a spectrophotometer (Uvikon 930, Kontron Instruments, Vienna, Austria).

### 2.3. Total Antioxidant Activity

The ABTS/H_2_0_2_/HRP method for food samples was used to determine the total antioxidant activity (TAA) [14], modified for animal samples [15]. The results are expressed in equivalents of mg Trolox mg^−1^ protein.

### 2.4. Lipid Peroxidation (LPO)

The amount of malondialdehyde (MDA) and 4-hydroxy-2(E)-nonenal (4-HNE) as an index of the oxidative destruction of lipids was measured with a lipid peroxidation kit from Calbiochem (No. 437634) [15]. Data are presented as nmol MDA+4-HNE/g protein.

### 2.5. Protein Damage

Protein carbonyl was determined spectrophotometrically at 340 nm (Uvikon 930) as previously described by Levine et al. [16] and modified by Coto-Montes and Hardeland [17]. The chromogen 2, 4-dinitrophenylhydrazine reacts with carbonyl groups of damaged proteins. Results are expressed as nmol of protein carbonyl mg^−1^ protein.

### 2.6. Proteasome Activity Assay

The 20S proteasome activity assay kit (Chemicon, CHEMICON International, Inc., Temecula, CA, USA) was used to measure proteasome activity. The assay detects fluorophore 7-amino-4-methylcoumarin (AMC) after cleavage from the labeled substrate LLVY-AMC by proteasomal chymotrypsin-like activity. Free AMC is detected by fluorometric quantification (380/460 nm). Data are expressed as Aminometilcoumarin units (AMC) mg^−1^ protein.

### 2.7. Cathepsin Activities

Cysteine proteinase cathepsin B (EC 3.4.22.1) was assayed fluorometrically (Cytofluor TM 2350, Millipore, Bedford, MA, USA) according to the method of Barret (1980) [18], with minor modifications for beef samples [19,20]. The aspartate proteinase, cathepsin D (EC 3.4.23.5), was assayed spectrophotometrically (Uvikon 930, Kontron Instruments, Milan, Italy) at 280 nm according to the procedure described by Takahashi and Tang (1981) [21], with minor modifications, using hemoglobin as the substrate [20]. The results are expressed as enzyme milliunits (mU) mg^−1^ protein for cathepsin B and as units (U) mg^−1^ protein for cathepsin D. The cathepsin B/D activity ratio was also calculated for each biological type and aging time.

### 2.8. Immunoblotting

Tissue samples were denatured in sample buffer at 100 °C for 5 min and separated by 12% SDS-PAGE at 100 V. After separation, proteins were transferred to PVDF membranes at 350 mA; Ponceau S staining was used to ensure equal loading. The membranes were blocked overnight in a cold room at 4 °C in 5% skim milk in Tris-Buffered Saline Tween-20 (TBS-T) buffer and then probed with antibodies against Beclin-1 (sc-10086) and β-actin (sc-1615) from Santa Cruz Biotechnology (Santa Cruz Biotechnology, Inc., Santa Cruz, CA, USA) and LC3 (PD014) from MBL (MBL Medical & Biological laboratories CO, Ltd., Nagoya, Japan), each previously diluted at 1:1000 in blocking buffer. After three 5-minute washes in TBS-T buffer, membranes were incubated with specific peroxidase-conjugated anti-immunoglobulin G (IgG, Singapore) secondary antibodies (Sigma-Aldrich, St. Louis, MO, USA) for 1 h at room temperature and washed twice for 20 min in TBS-T. The membrane was developed using the Western Blotting Luminol Reagent (Santa Cruz Biotechnology, sc-2048, Dallas, TX, USA) according to the manufacturer’s protocol. All of the data presented are representative of at least three separate experiments for each antibody. The protein abundances of all Western blots per condition were measured by densitometry of the bands on the films using ImageJ 2.9.0 open software (National Institutes of Health, Bethesda, MD, USA) and were normalized per amount of the loading control protein (β-actin) [15].

### 2.9. Sarcomeres

Longissimus dorsi pieces of 1 cm^3^ size were fixed in paraformaldehyde 4% and embedded in paraffin blocks. Sections of 8 μm were processed for hematoxylin and eosin (H&E) staining with established protocols [15]. Fibers in a longitudinal position were located and images were acquired at 40× magnification and 1300 × 1300 resolution, using a Nikon Coolpix 500 (Minato City, Tokyo, Japan) camera within the optical microscope Nikon Eclipse E4000. Sarcomeres were counted in the space of 20 μm using a fixed scale incorporated in the ADOBE PHOTOSHOP vs. 5.5. 15 measures in each slice. Data are expressed as the means of all replicates, expressed in microns.

### 2.10. Statistical Analysis

Data show mean ± standard error of mean (SEM) and are from a minimum of three independent experiments unless otherwise stated. All of the variables were analyzed with ANOVA considering biotype, aging time, and their interaction as the main effects. Once the interaction was probed, the effect of biotype or the effect of aging (with animal as the random factor) was tested. When significant, differences were analyzed by means of the Tukey post-hoc test (the Games-Howell test when variances were not homogeneous). The statistical analysis and graph representation was performed using the GraphPad Prism v8 software, which combines scientific graphing, statistics, and data organization.

## 3. Results

### 3.1. Antioxidant Status

The antioxidant status of the muscle cells showed important differences between biotypes, as shown in Figure 1.

#### 3.1.1. Total Antioxidant Activity (TAA)

Total antioxidant activity is considered a useful indicator of a system’s ability to regulate ROS-induced damage. TAA was lower in the (+/+) genotype, showing a significant decline at 24 h postmortem (*p* < 0.001). Thus, the normal genotype showed the lowest TAA over time and slightly exceeded heterozygote genotype at 2 h. However, the (*mh*/+) genotype did not show any changes in muscle TAA over time and, by 24 h, showed the greatest activity of all three biotypes. TAA values in the (*mh*/*mh*) genotype were the highest at 2 h (*p* < 0.01 and *p* < 0.0001 for (+/+) and (*mh*/+), respectively), but they also experienced a significant decline over time (*p* < 0.001) (Figure 1a).

#### 3.1.2. Lipid Peroxidation (LPO)

Lipid damage was assessed through LPO levels (Figure 1b). The three genotypes showed a similar pattern of lipid peroxidation with an increase over time. The (+/+) biotype showed a significant increase from 2 to 12 h postmortem (*p* < 0.01) and a stronger increase from 12 to 24 h (*p* < 0.0001). LPO levels in the (*mh*/+) genotype only exhibited a significant increase between 12 and 24 h (*p* < 0.0001 up 12 h). The (*mh*/*mh*) genotype showed the lowest increase, with significant differences observed only when comparing 2 and 24 h postmortem (*p* < 0.01). Despite these similar patterns in lipid damage, the heterozygote biotype had the highest values in the early hours, although the differences from homozygotes biotypes were not significant. At 24 h postmortem, all biotypes reached their peak in LPO levels, with the (+/+) genotype being the highest, followed by (only significant with (*mh*/*mh*), *p* < 0.0001), both also showing significant differences between them (*p* < 0.01) (Figure 1b).

#### 3.1.3. Protein Damage

The protein damage (PD) was expressed in terms of the protein carbonyl content, which informs about the oxidized protein. The three biotypes started with similar levels of PD. None of them showed significant differences as postmortem time increased, but the (+/+) and (*mh*/*mh*) biotypes showed higher values at 2 h postmortem, while heterozygous (*mh*/+) showed the highest values at 12 h (*p* < 0.05) and 24 h (*p* < 0.01).

### 3.2. Proteolytic Activities

#### 3.2.1. Proteosome Activity

It has been shown that the Ubiquitin–proteasome system degrades myofibrils and myofibrillar proteins [22]. So, we measured the 20S proteasome activity, and only the (+/+) biotype showed significant differences over maturation (Figure 2a). We observed a marked decrease in 20 s proteasome activity of the (+/+) biotype at 24 h compared to 12 h (*p* < 0.01). These lower levels of proteasome activity were significantly different compared to the heterozygous biotype (*p* < 0.05), which consistently maintained very similar levels. The (*mh*/*mh*) biotype showed a very similar pattern to the heterozygous (Figure 2a).

#### 3.2.2. Cathepsin Activities

Cathepsin activities were studied during the early postmortem period to understand the link between proteolytic activity and lysosomal viability (Figure 2b,c).

Cathepsin B activity was higher in the (*mh*/*mh*) biotype but significant differences were only found at 2 h with the heterozygous animals (*p* < 0.05). The (*mh*/+) biotype showed a significative rise in this enzymatic activity that peaked at 24 h (*p* < 0.05). Cathepsin D activity showed steady levels in all of the genotypes (Figure 2b). The highest activity was shown by (*mh*/*mh*), with significant differences compared to (+/+) at 2 h (*p* < 0.05) and to the (+/+) and (*mh*/+) biotypes at 24 h (*p* < 0.05). The cathepsin B/D activity ratio provides indirect information related to lysosomal viability. Higher levels of this ratio could be used as an indicator of pro-survival autophagy capability [9]. The only genotype that showed significant differences over time was the heterozygous one, which had its highest value at 24 h (Figure 2d; *p* < 0.0001 and *p* < 0.0003 for 2 and 12 h, respectively).

### 3.3. Sarcomeres

Sarcomere shortening may impact meat tenderness through different mechanisms [9,23]. In this study, sarcomere length was measured over the early postmortem period up to 24 h, instead of conducting a one-time measurement at 24 h. The (*mh*/*mh*) genotype presented longer sarcomeres in the longissimus dorsi muscle at 2 h after slaughter, although this difference was not statistically significant (Table 1). Moreover, this was the only biotype that showed a significant decrease in length between 2 and 24 h (*p* < 0.05).

To further understand the impact of time on the sarcomeres, we calculated the percentage of shortening relative to 2 h. Once again, the strongest shortening was observed in the (*mh*/*mh*) genotype compared to (*mh*/+) and (+/+) (*p* < 0.05 and *p* < 0.001, respectively).

### 3.4. Autophagy

Beclin 1, a class III phosphatidylinositol 3-kinase-interacting protein, plays an essential role in the early steps of autophagy during vesicle nucleation [24]. The abundance of Beclin 1 in (+/+) and the (*mh*/*mh*) biotypes revealed an opposite pattern during the early postmortem period (Figure 3a,b). In the (+/+) biotype, Beclin 1 increased over time, reaching its highest value at 24 h (*p* < 0.0001). In the (*mh*/*mh*) biotype, Beclin 1 showed the highest values at 2 and 12 h (*p* < 0.0001 and *p* < 0.0100, respectively) but decreased abruptly at 24 h (*p* < 0.0001) compared to the other two biotypes. Beclin 1 expression behavior in the heterozygous biotype resembled that observed in the (+/+) biotype, with a moderate increase at 24 h (*p* < 0.010) when it reached similar values to (*mh*/*mh*).

LC3 is the autophagosome ortholog of yeast Atg8 and its lipidation is essential for the execution of autophagy [25]. This modification allows the lipidated form (LC3-II) to bind to the autophagosome. Changes in this autophagosomal marker are helpful in understanding autophagic induction or activation [15]. LC3-II abundance in the (+/+) biotype showed an increase over time, with significant differences between 2 and 24 h (*p* < 0.05) and again between 12 and 24 h (*p* < 0.05). In the heterozygous biotype, we observed a similar pattern but without significant differences. The (*mh*/*mh*) biotype showed constant levels, with a higher abundance than the other biotypes at 2 h and lower at 24 h, but not presenting significant differences (Figure 3a,c).

## 4. Discussion

The quality label “Ternera Asturiana” ranks second in the protected geographical indication (PGI) market of fresh meat in Spain [3], with a significant economic impact on our region, Asturias. Numerous studies have focused on understanding the properties of the local breed “Asturiana de los Valles” and its beef quality [2,4,20]. This breed presents different biotypes with different postmortem tenderization patterns, being faster in *mh* genotypes [20]. The *mh* genotype also affects meat composition [4]. In fact, meat from (*mh*/*mh*) bulls has lower intramuscular fat and higher protein content than other beef breeds, making it more profitable [2,26]. However, how these characteristics from the *mh* genotype modulate redox balance and proteolysis, which directly impact meat tenderness, are still unknown.

Muscles from the (*mh*/*mh*) biotypes are characterized by a higher proportion of fast-twitch glycolytic muscle fibers, and therefore, they are considered to have a predominant glycolytic metabolism [27,28]. Muscles with glycolytic metabolism show a higher antioxidant response after training in rats [29]. Consequently, the elevated antioxidant response observed in (*mh*/*mh*) muscles during the early postmortem period (2 and 12 h) appears to be closely linked to the abundance of fast-twitch glycolytic fibers in this biotype.

This higher antioxidant activity may also explain the lower protein damage and lipid peroxidation observed in the muscle cells of the (*mh*/*mh*) biotype compared to the other groups at 24 h [29] The heterozygous biotype behaves more similarly to the (+/+) for TAA, but in terms of protein damage, it shows the highest levels, increasing over the studied postmortem period. This could be partly explained by the stability of the antioxidant activity during aging in this biotype, making it impossible to counteract the overproduction of free radicals resulting from anaerobic glycolytic metabolism, which may deeply impact the proteins of this heterozygous biotype, although the origin of these free radicals is not well characterized.

The ATP-independent 20S catalytic subunit of the proteasome, active in muscle, plays a pivotal role in the selective recognition and degradation of oxidized proteins [30]. Further, the role of the 20S proteasome in postmortem muscle has been associated with the loss of Z-disks in stored meat [31]. In agreement with this, our results showed a decrease in proteasome activity over postmortem time in homozygous biotypes, which may account for the Z-disks’ disintegration, while the heterozygous animals, which exhibited higher protein damage, maintained high activity at 24 h. Thus, the proteasome might be targeting the oxidized proteins accumulated in the muscle cells of this biotype.

The contribution of cathepsins to meat tenderization in the late stages of postmortem aging, when lysosomes are disrupted and pH conditions are favorable for acidic enzymes, has been previously described by our research group [20]. However, it is worth noting that lysosomal permeabilization and cathepsin release often constitute early events in cellular responses to oxidative stress [32]. In this study, cathepsins B and D exhibited higher activity in the *mh* genotypes, especially in the homozygous biotype, aligning with the meat tenderization observed for this biotype in previous studies [20]. Proteolysis is more pronounced in meat with longer sarcomeres, as proteases may have greater access through the filaments [23]. Thus, the longer sarcomeres of (*mh*/*mh*) may contribute to faster proteolysis compared to the other genotypes, leading to faster tenderization. In addition, the (*mh*/*mh*) biotype showed stronger sarcomere shortening. It has been described how the lack of myostatin causes more effective contraction [33], which may explain the shortening observed in this genotype.

Variations in the autophagic lysosomal pathway are directly related to the oxidative stress status of cells, playing a pivotal role in the initial stages of muscle conversion into meat and contributing to the early steps of meat tenderization [10]. Autophagic markers display a divergent pattern between the *mh* genotype and normal beef. Beclin 1, crucial for autophagosome maturation [24], shows an early peak in (*mh*/*mh*) beef compared to the normal biotype, suggesting either a faster activation of autophagy in response to oxidative stress after slaughter or increased basal autophagy. However, the autophagosome marker LC3-II remains stable during aging in the (*mh*/*mh*) biotype, which may indicate a delayed autophagic resolution [34]. Conversely, the (+/+) biotype shows a slower autophagic induction, with lower expression of Beclin 1 and LC3-II in the early stages (2 h postmortem) and an accumulation of autophagosomes toward the end of the studied period (24 h postmortem), when Beclin 1 and LC3-II expression are at their highest. Basal autophagy is responsible for maintaining cellular energy homeostasis by degrading damaged components, from proteins to organelles [35]. In contrast, induced autophagy is characterized by the degradation of intracellular macromolecules and organelles to meet energy requirements in response to stress such as situations of nutrient deficiency or high-energy demands, such as those of muscle after animals’ exsanguination [36]. In the present work, no significant differences in the cathepsin B/D activity ratio were found between biotypes, pointing to an active basal autophagy probably attempting to maintain cellular homeostasis [37]. In the heterozygous biotype, the B/D ratio increased over time, probably due to the higher lysosomal disruption in response to oxidative damage [9,38].

The observed differences in autophagy between genotypes, such as the delayed autophagic activation in normal biotypes, may be correlated with the muscles’ characteristics before slaughter. The (*mh*/*mh*) biotype may present intense basal autophagy to maintain healthy mitochondria, given the lower content in fast-twitch glycolytic fibers [39], while the (+/+) biotype shows a strong activation after stress [40,41]. A more in-depth analysis of certain cargo proteins, such as p62 or NBR1, or even organelles would be required to verify these observations [37].

## 5. Conclusions

In conclusion, muscular hypertrophy biotypes, particularly homozygous (*mh*/*mh*), showed higher antioxidant activity, increased lysosomal but not proteasomal proteolysis, and early autophagic activation in the muscle cells in early postmortem periods. Together, they may contribute to the earlier disintegration of the myofibers, which could explain why this biotype has a faster meat tenderization rate. In contrast, the normal biotype (+/+) exhibited slower but intense autophagic activation in the muscle cells in early postmortem stages and a slower meat tenderization rate. These results underline the relevance of early proteolysis, particularly autophagy, in modulating tenderization. We will analyze cargo proteins and different organelles in future studies to gain a better understanding of the role of autophagy during tenderization.

## Figures and Tables

**Figure 1 foods-13-00443-f001:**
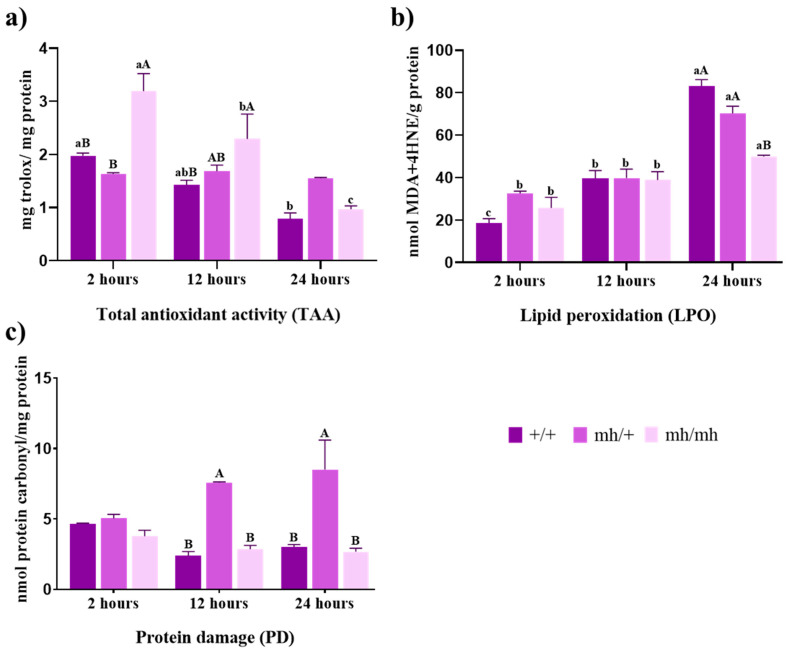
Antioxidant status. (**a**) Total antioxidant activity (TAA) expressed in mg Trolox mg ^−1^ protein; (**b**) lipid peroxidation (LPO) expressed in nmoles of MDA + 4HNE g^−1^ protein; (**c**) protein damage (PD) expressed in nmoles of carbonyl content mg^−1^ protein in meat from AV (+/+, *mh*/+, *mh*/*mh*) biotypes throughout the postmortem period studied (2 h to 24 h). For a biotype, lowercase letters indicate significant differences between times. For a given aging time, capital letters indicate significant differences between biotypes (*p* < 0.05).

**Figure 2 foods-13-00443-f002:**
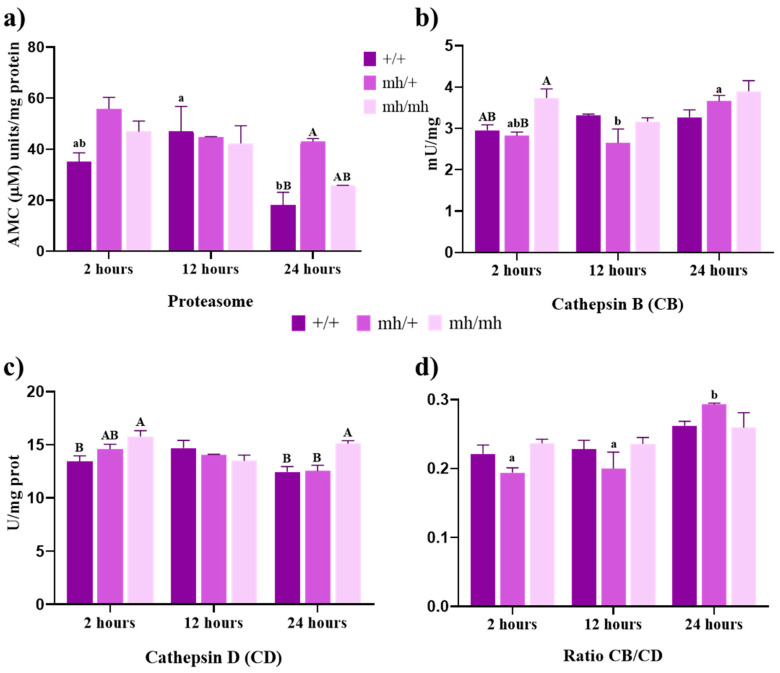
Proteolytic activities. (**a**) Proteasome activity expressed in arbitrary fluorescence units AMC mg^−1^ protein; (**b**) cathepsin B activity expressed in mU mg^−1^ protein; (**c**) cathepsin D activity expressed in U mg^−1^ protein; (**d**) ratio of cathepsin B/D (CB/CD) in meat from AV (+/+, *mh*/+, and *mh*/*mh*) biotypes throughout the postmortem period analyzed (2 h to 24 h). For a biotype, lowercase letters indicate significant differences between times. For a given aging time, capital letters indicate significant differences between biotypes (*p* < 0.05).

**Figure 3 foods-13-00443-f003:**
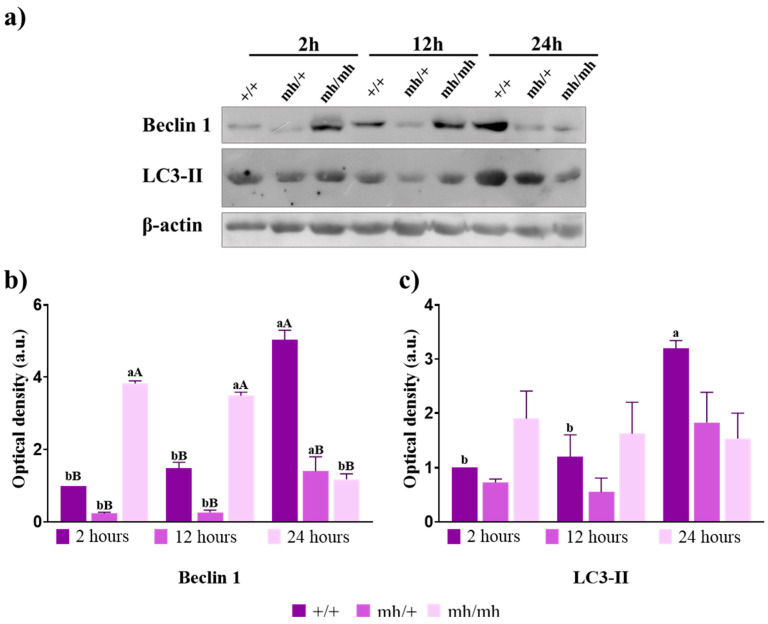
Autophagy markers. (**a**) Immunoblot analysis of Beclin 1, LC3-II, and β-actin in +/+, *mh*/+, and *mh*/*mh* biotypes′ meat throughout the postmortem period studied (2 h to 24 h). (**b**) Semi-quantitative optical density (arbitrary units) of Beclin 1 abundance normalized to β-actin. (**c**) Semi-quantitative optical density (arbitrary units) of LC3-II quantity normalized to β-actin. For a biotype, lowercase letters indicate significant differences between times. For a given aging time, capital letters indicate significant differences between biotypes (*p* < 0.05).

**Table 1 foods-13-00443-t001:** Sarcomere length. Data are expressed as mean ± standard deviation. Impact of biotype (columns) and aging time (rows) on the length of the sarcomeres. *p*-value is specified. % shortening means followed by letters (a and b) are significantly different. For a given aging time, capital letters indicate significant differences between biotypes (*p* < 0.05). NS means no significant difference.

		AV *(*+*/*+*)*	AV *(mh/*+*)*	AV *(mh/mh)*	
**Mean**	**2 h**	1.656 ± 0.044	1.621 ± 0.037	1.819 ± 0.025B	**NS**
	**12 h**	1.652 ± 0.050	1.661 ± 0.025	1.588 ± 0.078A	**NS**
	**24 h**	1.562 ± 0.027	1.596 ± 0.009	1.522 ± 0.125A	**NS**
**ANOVA**	**Sign. Time**	**NS**	**NS**	** *p* ** ** = 0.017**	
**% shortening**	**2–24 h**	5.587a	10.938a	16.637b	
**ANOVA**		**0.001**	**0.048**		

## Data Availability

Data are contained within the article.

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
