# Peer review of "Muscle Hypertrophy Is Linked to Changes in the Oxidative and Proteolytic Systems during Early Tenderization of the Spanish Breed “Asturiana de los Valles”"

_foods, 2024, doi:10.3390/foods13030443_

Round 1

Reviewer 1 Report

Comments and Suggestions for Authors

This research studied the changes in the oxidative and proteolytic systems during early tenderization of the Spanish breed “Asturiana de los Valles”. It is meaningful to investigate meat tenderization of livestock with different genotype, which provides a strategy on breeding. There are some comments to provide the manuscript quality.

1. Line 68: μ-calpain is one of the most important proteolysis system in postmortem meat. Why the authors did not evaluate its effect?

2. Line 169: It is inadequate to access meat tenderization by only referencing sarcomeres. Besides, the shear force and myofibril fragmentation index will better reflect meat tenderization.

3. Line 229: The date of 20S proteasome activity was very low. Please check.

4. Line 308-310: The TAA of mh/mh was higher at 2-12 h, but its lipid peroxidation was not significantly lower than other two groups. The discussion was less rigorous.

5. Line 331-332: Besides proteolysis, the sarcomeres also are related with muscle contraction. How does the authors think about this?

6. Line 365: The activity of 20S proteasome activity was lowest at 24 h in mh/mh. The author’s description was inconsistent with the results.

7. The significance of all figures are confusing. It is better to change a clearer sign. For example, the authors can use lowercase to represent the significance at different times in one group, and use the capital letter to represent the significance at the same time among different groups.

Comments on the Quality of English Language

Minor editing of English language required

Author Response

REVIEWER 1

Comments and Suggestions for Authors

This research studied the changes in the oxidative and proteolytic systems during early tenderization of the Spanish breed “Asturiana de los Valles”. It is meaningful to investigate meat tenderization of livestock with different genotype, which provides a strategy on breeding. There are some comments to provide the manuscript quality.

Thank you very much for taking the time to review this manuscript. Our group has numerous studies in this breed because the economic value of this meat is very important for our region. Thus, we have included the most relevant reports as suggested by the reviewer. Please find detailed responses below and the corresponding revisions/corrections highlighted/in track changes in the re-submitted files. We respond to your suggestions point by point:

  1. Line 68: μ-calpain is one of the most important proteolysis system in postmortem meat. Why the authors did not evaluate its effect?

Thank you so much for your question. We agree that μ-calpain is one of the most important proteolysis systems in postmortem meat. We checked that in our previous works with the Spanish breed “Asturiana de los Valles”[1]. In this work we sought how the presence of muscular hypertrophy produces faster exhaustion of μ-calpain activity. We have now included this study.

  1. Line 169: It is inadequate to access meat tenderization by only referencing sarcomeres. Besides, the shear force and myofibril fragmentation index will better reflect meat tenderization.

Thank you so much for your point and indeed, sarcomeres are not enough to describe meat tenderization. We never intended to make such strong claim, so we have toned down our allegations. Besides, “Asturiana de los Valles” meat tenderization have been well characterized previously by our group and others. Particularly, we have studied the shear force from this breed in[1,2] , where we performed the Warner–Bratzler (WB) shear test. The myofibril fragmentation index was shown in[1] (1).

  1. Line 229: The date of 20S proteasome activity was very low. Please check.

Thank you so much for pointing this out. We have corrected it.

  1. Line 308-310: The TAA of mh/mh was higher at 2-12 h, but its lipid peroxidation was not significantly lower than other two groups. The discussion was less rigorous.

You are right in the assessment that the lipid peroxidation is not lower while the TAA is higher. We believe that the higher TAA activity in the mh/mh causes the lower lipid peroxidation at 24 hours as it could reduce the reactive species that produce the lipid peroxidation. We have improved our discussion to reflect these considerations.

  1. Line 331-332: Besides proteolysis, the sarcomeres also are related with muscle contraction. How does the authors think about this?

Thank you so much for this question. The sarcomeres are the basic contractile unit of the muscle as you mention. It has been described how the lack of myostatin causes a more effective contraction and relaxation of the muscle as well as an increase of the type IIb fibers (glycolytic ones) with a mitochondrial depletion [3]. This has been tested in mice, and its relevance to human physical performance has also been analyzed[4]. Myostatin mutation in “Asturiana de los Valles” could be the cause of the strong sarcomere shortening in mh/mh genotype. We include this in the discussion.

  1. Line 365: The activity of 20S proteasome activity was lowest at 24 h in mh/mh. The author’s description was inconsistent with the results.

Thank you, it is corrected.

  1. The significance of all figures are confusing. It is better to change a clearer sign. For example, the authors can use lowercase to represent the significance at different times in one group, and use the capital letter to represent the significance at the same time among different groups.

It is now changed.

REFERENCES

  1. Sierra, V.; Guerrero, L.; Fernandez-Suarez, V.; Martinez, A.; Castro, P.; Osoro, K.; Rodriguez-Colunga, M.J.; Coto-Montes, A.; Olivan, M. Eating quality of beef from biotypes included in the PGI "Ternera Asturiana" showing distinct physicochemical characteristics and tenderization pattern. Meat Sci 2010, 86, 343-351, doi:10.1016/j.meatsci.2010.05.007.
  2. Beatriz Caballero, V.S., Mamen Oliván, Ignacio Vega-Naredo, Cristina Tomás-Zapico, Óscar Alvarez-García, Delio Tolivia, Rüdiger Hardeland, María Josefa Rodríguez-Colunga, Ana Coto-Montes. Activity of cathepsins during beef aging related to mutations in the myostatin gene. Journal of the Science of Food and Agriculture 2007, 87, 192 – 199, doi: https://doi.org/10.1002/jsfa.2683.
  3. Amthor, H.; Macharia, R.; Navarrete, R.; Schuelke, M.; Brown, S.C.; Otto, A.; Voit, T.; Muntoni, F.; Vrbova, G.; Partridge, T.; et al. Lack of myostatin results in excessive muscle growth but impaired force generation. Proc Natl Acad Sci U S A 2007, 104, 1835-1840, doi:10.1073/pnas.0604893104.
  4. Gineviciene, V.; Jakaitiene, A.; Pranckeviciene, E.; Milasius, K.; Utkus, A. Variants in the Myostatin Gene and Physical Performance Phenotype of Elite Athletes. Genes (Basel) 2021, 12, doi:10.3390/genes12050757.

Reviewer 2 Report

Comments and Suggestions for Authors

There are many revisions that must be done, as illustrated in the pdf comments.

Author Response

Thank you very much for taking the time to review this manuscript. Please find detailed responses in the attached pdf as we were answering your questions in situ and the corresponding revisions/corrections highlighted/in track changes in the re-submitted files.

Reviewer 3 Report

Comments and Suggestions for Authors

The article with the title "Muscle hypertrophy is linked to changes in the oxidative and proteolytic systems during early tenderization of the Spanish breed “Asturiana de los Valles” has 14 pages, 45 references, it has three split images (each composed of three sub-images) and 1 table.

I have some comments on the formal facts of the manuscript and the content part of the experiment.

The entire article needs to be proofread, e.g. the fact that something was followed post-mortem is stated in italics. However, it applies to more cases.

Lines 35-37: Changes in the redox balance of muscle cells have more relevance in the first hours after slaughter and total antioxidant activity seems to follow the shortening of the sarcomeres, an important pa- 36 rameter to understand meat tenderness."

have more relevance - were it significant?

Lines 40-42: "Autophagic activation showed significant differences ..."

significant differences - and what about P, which P<0.0? 

Lines 85-88: Eighteen yearling bulls of a local breed from Northern Spain, AV, were studied. This 85 is a meat-specialized breed and presents a myostatin mutation (mh) that induces doublemuscling character, giving rise to three different biotypes: homozygous (mh/mh), heterozygous (mh/+) and normal (+/+).

- local breed - ok, but for this breed it would be better to describe why the experiment is so important for it/us/readers, what the experiment is supposed to contribute to and give a basic definition of the material, either in sentences in the Introduction or by referring to other studies with a basic explanation in the Methodology of the experiment.

Line 89: "At three months of age..."

- This is quite vague, even if it was done randomly and just for the sake of an experiment, or if it happens regularly. It should be expressed more specifically and in days.

Line 262: According to the instructions on how to write the manuscript, the title of Table 1 should be at the top and not below this table.

The information in the Results section is pretty shallow and the explanation in the discussion doesn't help much. The experiment was carried out, it is very briefly discussed here in the article, but I cannot get rid of the impression that the manuscript could be better developed, especially by linking the results with a discussion and explaining why the authors investigated this and what is the benefit and for whom.

Author Response

Thank you very much for taking the time to review this manuscript. We have proofread the manuscript, thank you so much for noticing that. Please find the detailed responses below and the corresponding revisions/corrections highlighted/in track changes in the re-submitted files. We will respond to your suggestions point by point:

-Lines 35-37: Changes in the redox balance of muscle cells have more relevance in the first hours after slaughter and total antioxidant activity seems to follow the shortening of the sarcomeres, an important pa- 36 rameter to understand meat tenderness."

have more relevance - were it significant?

To clarify the text, we have remodeled that part of the abstract.

Lines 40-42: "Autophagic activation showed significant differences ..."

significant differences - and what about P, which P<0.0? 

We have included the p-value.

Lines 85-88: Eighteen yearling bulls of a local breed from Northern Spain, AV, were studied. This 85 is a meat-specialized breed and presents a myostatin mutation (mh) that induces doublemuscling character, giving rise to three different biotypes: homozygous (mh/mh), heterozygous (mh/+) and normal (+/+).

- local breed - ok, but for this breed it would be better to describe why the experiment is so important for it/us/readers, what the experiment is supposed to contribute to and give a basic definition of the material, either in sentences in the Introduction or by referring to other studies with a basic explanation in the Methodology of the experiment.

Thank you for your question. How the lack of myostatin affects muscle biology has been studied before. It has also been studied how autophagy changes in the case of myostatin mutations. These studies focused on muscle performance using mice models and then in different human samples. However, the effect of the mh-mutation in autophagy during meat aging is unknown. This knowledge will help to improve customers’ satisfaction. We have included this in the introduction.

Line 89: "At three months of age..."

- This is quite vague, even if it was done randomly and just for the sake of an experiment, or if it happens regularly. It should be expressed more specifically and in days.

It has been changed, thank you.

Line 262: According to the instructions on how to write the manuscript, the title of Table 1 should be at the top and not below this table.

Thank you so much for that, we didn’t realize, now it has been changed.

The information in the Results section is pretty shallow and the explanation in the discussion doesn't help much. The experiment was carried out, it is very briefly discussed here in the article, but I cannot get rid of the impression that the manuscript could be better developed, especially by linking the results with a discussion and explaining why the authors investigated this and what is the benefit and for whom.

Thank you so much for your kind words, we have tried to improve the results and the discussion to answer these concerns.

Round 2

Reviewer 3 Report

Comments and Suggestions for Authors

After reading the corrected manuscript, I note that I have no further comments.

The manuscript is corrected in terms of comments, so I can recommend its further process leading to its publication.